# Poly(3-mercapto-2-methylpropionate), a Novel α-Methylated Bio-Polythioester with Rubber-like Elasticity, and Its Copolymer with 3-hydroxybutyrate: Biosynthesis and Characterization

**DOI:** 10.3390/bioengineering9050228

**Published:** 2022-05-23

**Authors:** Lucas Vinicius Santini Ceneviva, Maierwufu Mierzati, Yuki Miyahara, Christopher T. Nomura, Seiichi Taguchi, Hideki Abe, Takeharu Tsuge

**Affiliations:** 1Department of Materials Science and Engineering, Tokyo Institute of Technology, 4259 Nagatsuta, Midori-ku, Yokohama 226-8502, Japan; santini.l.aa@m.titech.ac.jp (L.V.S.C.); mirzat.m.aa@m.titech.ac.jp (M.M.); miyahara.y.aa@m.titech.ac.jp (Y.M.); 2Department of Biological Sciences, College of Science, University of Idaho, 875 Perimeter Dr., Moscow, ID 83844-3010, USA; ctnomura@uidaho.edu; 3Graduate School of Science, Technology and Innovation, Kobe University, 1-1 Rokkodai-cho, Nada, Kobe 657-8501, Japan; staguchi86@people.kobe-u.ac.jp; 4Bioplastic Research Team, RIKEN Center for Sustainable Resource Science, 2-1 Hirosawa, Wako, Saitama 351-0198, Japan; habe@riken.jp

**Keywords:** polythioester, polyhydroxyalkanoate, 3-hydroxybutyrate, bioplastic, biopolymer, alpha-methylated, rubber-like elasticity

## Abstract

A new polythioester (PTE), poly(3-mercapto-2-methylpropionate) [P(3M2MP)], and its copolymer with 3-hydroxybutyrate (3HB) were successfully biosynthesized from 3-mercapto-2-methylpropionic acid as a structurally-related precursor. This is the fourth PTE of biological origin and the first to be α-methylated. P(3M2MP) was biosynthesized using an engineered *Escherichia coli* LSBJ, which has a high molecular weight, amorphous structure, and elastomeric properties, reaching 2600% elongation at break. P(3HB-*co*-3M2MP) copolymers were synthesized by expressing 3HB-supplying enzymes. The copolymers were produced with high content in the cells and showed a high 3M2MP unit incorporation of up to 77.2 wt% and 54.8 mol%, respectively. As the 3M2MP fraction in the copolymer increased, the molecular weight decreased and the polymers became softer, more flexible, and less crystalline, with lower glass transition temperatures and higher elongations at break. The properties of this PTE were distinct from those of previously biosynthesized PTEs, indicating that the range of material properties can be further expanded by introducing α-methylated thioester monomers.

## 1. Introduction

Global plastic production reached 367 million tons by 2020 [1]. Between 1950 and 2015, only 9% of plastic waste was recycled, with 12% incinerated and 79% accumulated in landfills and the environment [2]. There is increasing concern regarding plastic waste pollution, particularly in oceans, as 80% of these plastics are of land origin [3]. Thus, it is projected that by 2050, there will be more plastics than fish in the oceans [4].

The development of biodegradable and bio-based plastics, known as “bioplastics”, has gained attention [5]. Polyhydroxyalkanoates (PHAs) are a family of aliphatic polyesters naturally produced from sugars, fatty acids, and amino acids by bacteria as intracellular energy reserves in the presence of excess carbon sources and limited nutrients [6,7]. Bioplastics exhibit intrinsic biodegradability and biocompatibility in any medium or marine environment [8,9,10]. PHAs also exhibit extraordinary versatility, as polyesters from more than 160 different monomers can be produced by wild-type or engineered bacteria from structurally related or unrelated carbon sources [11].

Poly(3-hydroxybutyrate) [P(3HB)], the most common PHA in natural PHA-producing bacteria [6], is stiff and brittle [12] and has low thermal stability [13]; numerous methods have been investigated to improve its properties, including thermal processing, blending, fiber inclusion, and copolymerization [14]. Most copolymerization approaches have focused on the biosynthesis of 3HB-based copolymers with medium-chain length monomers, such as 3-hydroxyhexanoate, as their elastomeric characteristics can result in a softer and more flexible structure than those of P(3HB) [15] or PHA monomers containing shorter chain lengths (i.e., 2-hydroxyalkanoate monomers), fluorine, or ring-structures [16,17,18]. Recently, the α-methylated structure of poly(3-hydroxy-2-methylbutyrate) [P(3H2MB)] was used to fabricate PHA materials with high flexibility and superior thermal stability because of its fast crystallization behavior and highest melting temperature (*T_m_*) among PHAs [19].

Polythioesters (PTEs) were first chemically synthesized approximately 70 years ago [20]; however, they have never been produced on an industrial scale and established commercially because of their high production costs, low yields, and use of toxic substances [21]. In 2001, by using structurally related thioester precursors and other carbon sources, Lütke-Eversloh et al. [22] identified that a copolymer of 3HB and 3-mercaptopropionate (3MP) could be biosynthesized by *Ralstonia eutropha* H16 via its inherent P(3HB) synthesis pathway, making it the eighth class of biological polymers. In sequence, copolymers of 3HB with 3-mercaptobutyrate (3MB) [23] and 3-mercaptovalerate (3MV) [24] were synthesized from their structurally related precursors in *R. eutropha*. Notably, their homopolymers, P(3MP), P(3MB) and P(3MV), could be biosynthesized by constructing a recombinant strain of *Escherichia coli* via a non-natural pathway containing butyrate kinase to phosphorylate 3-mercaptoalkanoate precursors, phosphotransbutyrylase to convert 3-mercaptoalkanoate-Pi esters to 3-mercaptoalkanoate-CoA thioesters, and PHA synthase to polymerize these thioesters by releasing CoA [25]. Comparison of the properties of biosynthesized PTE homopolymers and those of their structurally analogous oxyester PHAs indicated that they can have a higher glass transition temperature (*T_g_*) and/or higher *T_m_* than their oxygen analogs [25]. PTE homopolymers may also have higher elongations at break, as copolymers of 3MP with 3HB showed higher elasticity than those of 3HB with an equivalent oxyester [26].

In this study, we biosynthesized a new PTE homopolymer and its copolymers with 3HB by an engineered *E. coli* and using 3-mercapto-2-methylpropionic acid as the thioester precursor, making it the fourth PTE to be biosynthesized and the first α-methylated PTE. P(3M2MP) and its copolymers had their chemical structure confirmed by nuclear magnetic resonance (NMR), and their thermal and mechanical properties were also investigated by differential scanning calorimetry (DSC) and tensile tests.

## 2. Materials and Methods

### 2.1. Bacterial Strain and Plasmids

The host strain, plasmids, and biosynthesis strategies used in this study were the same as in the previous studies [19,27]. The host strain for PHA/PTE accumulation was *E. coli* LSBJ, a *fadB* and *fadJ* knockout mutant of *E. coli* LS5218, which enables control of the repeating unit composition in PHA biosynthesis [28]. For P(3HB-*co*-3M2MP) biosynthesis, two plasmids were introduced into the host strains, pTTQ19PCT and pBBR1phaP(D4N)CJ_Ac_AB_Re_ NSDG. pTTQ19PCT contains the propionyl-CoA transferase (PCT) gene from *Megasphaera elsdenii*, which supplies 3M2MP-CoA, whereas pBBR1phaP(D4N)CJ_Ac_AB_Re_ NSDG contains the phasin (PhaP_Ac_) gene from *Aeromonas caviae* with the point mutation D4N that enables high PHA accumulation by enhancing the expression of the *phaPCJ* operon [29], PHA synthase (PhaC_Ac_) gene from *A. caviae* with N149S and D171G point mutations that enhance incorporation of the second monomer unit of the 3HB-based copolymer [30], (*R*)-specific enoyl-CoA hydratase gene (*phaJ*_Ac_), and genes for the enzymes, 3-ketothiolase (PhaA_Re_) and acetoacetyl-CoA reductase (PhaB_Re_) from *R. eutropha* H16 that supply the 3HB precursor for PhaC_Ac_ polymerization. For P(3M2MP) homopolymer biosynthesis, the plasmid pTTQ19PCT was also introduced, while the other plasmid had the deletion of the *phaAB*_Re_ and *phaJ*_Ac_ genes, becoming the plasmid pBBR1phaP(D4N)CJ_Ac_ NSDG. To obtain pBBR1phaP(D4N)C_Ac_ NSDG, a 7-kb DNA fragment was amplified from pBBR1phaP(D4N)CJ_Ac_AB_Re_ NSDG using PCR primer sets (5′-gttgggcaggcaaacacggggtt-3′ and 5′-gatccactagttctagagcggcc-3′). The resulting PCR fragment was treated with the Mighty Cloning Reagent Set Blunt End Kit (Takara Bio, Shiga, Japan) and self-ligated using a DNA Ligation Kit (Takara). Figure 1 illustrates the proposed biosynthetic pathways for P(3HB-*co*-3M2MP) and P(3M2MP).

### 2.2. P(3HB-co-3M2MP) and P(3M2MP) Biosynthesis, Harvest, and Polymer Content

For copolymer production, recombinant *E. coli* LSBJ was pre-cultivated for 18 h at 30 °C with reciprocal shaking at 160 rpm in a 50 mL flask containing 20 mL of lysogeny broth (LB) medium (i.e., 10 g NaCl, 10 g tryptone, and 5 g bacto-yeast extract per liter of distilled water) with 50 mg/L kanamycin and 50 mg/L carbenicillin. Thereafter, 10 mL/L of pre-culture was inoculated into 500 mL shake flasks containing approximately 100 mL of M9-modified medium (17.1 g/L Na_2_HPO_4_∙12H_2_O, 3 g/L KH_2_PO_4_, 2.5 g/L bacto-yeast extract, 0.5 g/L NaCl) containing 2 mL/L of 1 M MgSO_4_∙7H_2_O, 0.1 mL/L of 1 M CaCl_2_, 50 mg/L kanamycin, 50 mg/L carbenicillin, 20 g/L glucose, and different concentrations of (*R*,*S*)-3-mercapto-2-methylpropionic acid (Tokyo Kasei Kogyo Co., Tokyo, Japan; 0.25–2.5 g/L) as a 3M2MP precursor, with the pH adjusted to 7.0 using NaOH. Gene expression was induced by adding 1 mM isopropyl β-D-1-thiogalactopyranoside (IPTG). The culture for P(3HB-*co*-3M2MP) production was maintained for 72 h at 30 °C with reciprocal shaking at 130 rpm.

Based on the results of Watanabe et al. [27] and Furutate et al. [19], several changes were made in the concentrations of the culture medium used for P(3M2MP) homopolymer biosynthesis. First, an additional pre-cultivation step was performed, in which recombinant *E. coli* LSBJ was inoculated into 1.7 mL of LB medium containing 50 mg/L kanamycin and 50 mg/L carbenicillin in 5 mL test tubes for 4 h at 37 °C with reciprocal shaking at 160 rpm. This seed culture (0.2 mL) was further inoculated into a 50 mL flask containing 20 mL LB medium, 50 mg/L kanamycin, and 50 mg/L carbenicillin for an additional 18 h at 30 °C with reciprocal shaking at 160 rpm. The pre-culture (5 mL) was inoculated into 500 mL shake flasks containing approximately 100 mL of M9-modified medium with 2 mL/L 1 M MgSO_4_∙7H_2_O, 0.1, 1 M CaCl_2_, 50 mg/L kanamycin, and 50 mg/L carbenicillin, and grown without precursors and IPTG for 4 h at 30 °C with reciprocal shaking at 130 rpm. Thereafter, 3.75 g/L glucose, 1.2 g/L pH-neutralized 3-mercapto-2-methylpropionic acid, and 1 mM IPTG were added to the culture medium, and cultivation was continued until a total biosynthesis time of 76 h was reached.

After cultivation, the cells were harvested by centrifugation, washed twice with distilled water, and lyophilized for 72 h in previously weighed tubes to obtain the dry cell weight. The polymer content was determined by ultrasonication extraction as described previously with some modifications [31]. Approximately 100 mg of lyophilized cells was added to 50 mL plastic tubes and resuspended in 20 mL of distilled water. Subsequently, 13 mL 10% sodium dodecyl sulfate (SDS) was added to a final concentration of approximately 4% SDS. This solution was ultrasonicated continuously for 4 min at an output level of 4 (15 W), harvested by centrifugation with three rinsing steps (20 mL distilled water, 5 mL methanol, and 20 mL distilled water), and lyophilized for 24 h to determine the polymer content.

To produce sufficient polymers for characterization, biosynthesis was performed at a larger scale in 2 L shake flasks containing 1 L M9-modified medium at the same concentrations as described above, except that the reciprocal shaking speed was adjusted to 103 rpm. For the second step of pre-cultivation, the scale was increased to 500 mL shake flasks containing 100 mL of LB medium with 50 mg/L kanamycin and 50 mg/L carbenicillin, and the reciprocal shaking speed was adjusted to 130 rpm.

### 2.3. Polymer Film Preparation

The polymers produced on a larger scale were harvested by centrifugation, lyophilized for 72 h, extracted with chloroform for 72 h at room temperature, and purified by precipitation with methanol. The purified polymers were dissolved in chloroform to prepare the polymer films. The polymer solution was filtered once or twice with 0.45 µm polytetrafluoroethylene filter membranes and added to a perfluoroalkoxy alkane (PFA) Petri dish with an internal diameter of approximately 7.7 cm. After solvent evaporation, the polymer films were aged for at least three weeks at room temperature before characterizing their thermal and mechanical properties.

### 2.4. Polymer Structure Characterization

For one-dimensional ^1^H NMR, two-dimensional ^1^H-^1^H correlation spectroscopy (^1^H-^1^H COSY), and ^1^H-^13^C heteronuclear single quantum coherence (^1^H-^13^C HSQC) NMR, 10–20 mg of the purified and filtered polymers was dissolved in 1 mL of CDCl_3_ and filtered through 0.45 µm polyvinylidene fluoride filter membranes. In contrast, for one-dimensional ^13^C NMR, 20-40 mg was dissolved in 1 mL of CDCl_3_ and filtered through 0.45 µm polyvinylidene fluoride filter membranes (NMR; 400 Hz; BioSpin AVANCE III 400A, or 500 MHz AVANCE III HD with CryoProbe, Bruker, Billerica, MA, USA). The chemical structure was determined using ^1^H and ^13^C NMR and confirmed using ^1^H-^1^H COSY and ^1^H-^13^C HSQC. The 3M2MP unit content of the P(3HB-*co*-3M2MP) copolymers was determined by integrating the methine (>CH−) peaks in the ^1^H NMR spectra. The sequence distribution was investigated by calculating the *D* value [32] by integrating the carbonyl groups in the ^13^C NMR spectra, and the peaks for the four possible combinations of the copolymer units were assigned using ^1^H-^13^C heteronuclear multiple bond correlation (^1^H-^13^C HMBC). Finally, Fourier transform infrared (FTIR) spectroscopy using an FT/IR-4600 spectrometer (Jasco, Tokyo, Japan) with attenuated total reflection (ATR) (model ATR PRO400-S, Jasco, Tokyo, Japan) was performed to confirm the presence of thioester chemical groups in the polymer molecule.

### 2.5. Molecular Weight

For molecular weight determination, 2–3 mg of the polymer samples were dissolved in HPLC-grade chloroform at 1 mg/mL, filtered through 0.45 µm polyvinylidene fluoride filter membranes, and analyzed using gel permeation chromatography (GPC) on a Shimadzu Nexera 40 GPC system (Kyoto, Japan) with a Shodex RI-504 refractive index detector (Shanghai, China). The gel permeation chromatography system was operated in a column oven at 40 °C. HPLC-grade chloroform was used as the eluent at a flow rate of 0.3 mL/min. Each sample was analyzed for 28 min. Calibration curves were plotted using polystyrene standards with low polydispersity.

### 2.6. Thermal Properties

The thermal properties of the polymers were analyzed using DSC 8500 (Perkin-Elmer, Waltham, MA, USA) under a nitrogen atmosphere. Each sample (5–7 mg) was encapsulated in an aluminum pan and subjected to two heating steps. The samples were heated from −50 °C to 200 °C at 20 °C/min (1st heating scan). The melted samples were heated at 200 °C for 1 min, rapidly decreased to −50 °C, and then heated from −50 °C to 200 °C at 20 °C/min (2nd heating scan). The *T_m_* and enthalpy of fusion (*ΔH_m_*) were determined from the 1st heating scan curve, whereas the *T_g_* and cold crystallization temperature (*T_cc_*) were determined from the 2nd heating scan curve.

### 2.7. Mechanical Properties

The tensile strength, yield strength, Young’s modulus, and elongation at break of the polymers were determined using the stress–strain curves measured with a Shimadzu EZ-S 500N testing machine at a strain rate of 5 mm/min. The samples were dumbbell-shaped using a super dumbbell cutter (SDMP-1000, ISO 37-4/ISO 527-2-5B) with a gauge length of 10 mm, width of 2 mm, and thickness of approximately 100 µm.

## 3. Results

### 3.1. P(3HB-co-3M2MP) and P(3M2MP) Biosynthesis and Polymer Content

Copolymers of P(3HB-*co*-3M2MP) were biosynthesized by recombinant *E. coli* LSBJ containing the plasmids pTTQ19PCT (for *pct* expression) and pBBR1phaP(D4N)JC_Ac_AB_Re_ NSDG (for *phaPCJ* and *phaAB* expression) with glucose and increasing concentrations of 3-mercapto-2-methylpropionic acid (0.25–2.5 g/L) as carbon source and 3M2MP precursor, respectively. Use of 0.25–1.5 g/L of 3-mercapto-2-methylpropionic acid led to stable cell growth and a polymer content of approximately 3.5 g/L and 68 wt%, respectively. The 3M2MP content also increased from 5.5 mol% to 53.9 mol%, as summarized in Table 1. However, when the concentrations exceeded 2.0 g/L, cell growth and the polymer content were significantly increased, with ranges of 4.29–5.06 g/L and 72.1–77.2 wt%, respectively. Notably, the 3M2MP content reached a maximum of 54.8% at 2 g/L and then decreased sharply to 10.7 mol% at 2.5 g/L.

The P(3M2MP) homopolymer was biosynthesized by recombinant *E. coli* LSBJ containing the plasmids pTTQ19PCT and pBBR1phaP(D4N)C_Ac_ NSDG cultured with 3.75 g/L of glucose and 1.2 g/L of 3-mercapto-2-methylpropionic acid. As expected, the cell growth and polymer content of P(3M2MP) were significantly lower than those of the 3HB-based copolymer, with 1.28 g/L of cell growth and 8.4 wt% of P(3M2MP) content.

### 3.2. Chemical Structure and Sequence Distribution Characterization

Based on ^1^H NMR analysis of the P(3M2MP) homopolymer, four peaks directly associated with its structure, A, B, and C, were identified with peak integrations of 0.99–1.00, at 3.14, 3.01, and 2.86 ppm, and D with a peak integration of 3.16 at 1.25 ppm. Other significant peaks at 7.26, 1.58, and 0 ppm corresponded to the solvent *d*-chloroform, water moisture, and internal standard tetramethylsilane, respectively. ^1^H-^1^H COSY NMR revealed correlations between A and B and between C and D. Based on this result and those of ^1^H NMR integration, A and B were identified as the two protons of the methylene group, C was the proton of the methine group, and D was the proton of the methyl group. Using ^13^C NMR, peaks E, F, G, and H at 201.3, 48.4, 31.6, and 17.6 ppm, respectively, were found to be directly associated with its structure, which was confirmed using ^1^H-^13^C HSQC NMR. This result indicates that A and B correlate with G, serving as the methylene group, C correlates with F as the methine group, D correlates with H as the methyl group, and E serves as the protonless carbonyl group, confirming the expected structure of the homopolymer structurally related to 3-mercapto-2-methylpropionic acid. Figure 2a shows the ^1^H NMR, Figure 2b ^1^H-^1^H COSY NMR, Figure 2c ^13^C NMR, and Figure 2d ^1^H-^13^C HSQC NMR spectra. Moreover, based on the ATR-FTIR results shown in Figure 3, the thioester group was confirmed based on the peaks at 1682 and 956 cm^−1^, which were associated with dialkyl thioester carbonyl stretching (1700–1680 cm^−1^) and dialkyl thioester C-S stretching (1035–930 cm^−1^), respectively [33].

Based on ^1^H NMR analysis of the copolymers P(3HB-*co*-3M2MP) in Figure 4, the 3HB chemical shifts were similar to those found by Lütke-Eversloh et al. [22] for the copolymer, P(3HB-*co*-3MP). However, because of the presence of a methyl group in the 3M2MP unit, overlap with the methyl group of 3HB was observed. Therefore, the comonomer composition was estimated by integrating the methine groups.

Based on the sequence distribution of the polymer, as a copolymer with two different units, four types of unit combinations can occur, 3HB-3M2MP, 3HB-3HB, 3M2MP-3M2MP, and 3M2MP-3HB, with the carbonyl group split into four peaks using a Bernoullian statistical model. The sequence distribution was predicted from the parameter “*D*” [32] as follows. For simplification, 3HB is expressed as “*O*” and 3M2MP as “*S*”.
(1)$$D=\frac{F_{OO}\cdot F_{SS}}{F_{OS}\cdot F_{SO}}\;$$
where *F_XY_* is the mole fraction of the carbonyl group in the *XY* sequence in the ^13^C NMR spectrum. Therefore, when *D* = 1, the copolymer is statistically random. However, when *D* is markedly larger than 1, the copolymer is blocky, and when it is markedly smaller than 1, it is an alternating copolymer [32].

The ^13^C NMR spectra of the copolymer samples showed two extra peaks 195.2 and 173.5 ppm that were not present in the spectra of their homopolymers. Thereafter, based on the ^1^H-^13^C HMBC for the copolymer samples, these peaks were found to interact with the protons of the 3M2MP and 3HB units. Accordingly, these are the carbonyl peaks of 3M2MP-3HB and 3HB-3M2MP diad, respectively. Figure 5a shows the ^13^C NMR spectrum and Figure 5b the ^1^H-^13^C HMBC spectrum of Sample 5. The *D* parameter was calculated for the copolymer samples using the carbonyl peaks detected in the ^13^C NMR spectra. Table 2 presents the results for the *F_XY_* and *D* parameters for each sample and shows that all samples had *D* values significantly greater than 1, indicating that they have blocky sequence distributions. When the amount of the 3M2MP precursor added to the medium was small, the degree of the block sequence tended to be high.

### 3.3. Molecular Weight and Thermal Properties

The weight-average molecular weight (*M_w_*) of the samples was considered high, varying from 7.7 × 10^5^ to 17.5 × 10^5^ g/mol. The values are in the upper limit of the 1 to 20 × 10^5^ g/mol range commonly found for biosynthesized P(3HB) [34]. In contrast, other PTE and PTE copolymers with 3HB have markedly smaller *M_w_* and *M_w_*/*M_n_*, as reported previously [22,23,24,25,26]. Moreover, except for in Sample 4, decreasing *M_w_* and increasing *M_w_*/*M_n_* trends were observed, possibly because of the increasing amounts of unreacted 3-mercapto-2-methylpropionic acid; at increasing dosage concentrations, this reagent acted as a chain-transfer agent, ultimately decreasing the average molecular weight of the polymer and increasing the number of synthesized chains because of its chain termination function [34].

Based on the thermal properties, the homopolymer (Sample 7) displayed clear amorphous behavior, with a negative *T_g_* and no melting or cold crystallization peaks. Consequently, in Samples 1–3, with increasing amounts of 3M2MP incorporated into the polymer structure, *T_g_* and *ΔH_m_* decreased until only *T_g_* remained, resulting in amorphous characteristics for Sample 4. Samples 5 and 6 exhibited different behaviors. The 3M2MP unit content of Sample 5 was similar to that of Sample 4 but was not amorphous. Sample 5 had two *T_g_* peaks with lower *ΔH_m_* and *T_cc_* values but a higher *T_m_* compared with those of Sample 3. This finding may be related to differences in its microstructure compared with that of Sample 4. The *T_g_* and *T_cc_* of Sample 6 were similar to those of Sample 1; however, this sample exhibited a higher *T_m_* and *ΔH_m_*, possibly because of its higher crystallinity, higher *M_w_* and *M_w_*/*M_n_*, and lower 3M2MP fraction, despite the higher precursor addition. Moreover, as there are no published reports of biosynthesis and material property of its equivalent oxoester polymer [35,36], we could not confirm the high thermal stability of P(3M2MP). However, compared with other bio-PTE homopolymers, P(3M2MP) is the first amorphous PTE [25].

Table 3 shows the results of the molecular weight and thermal property analyses of the samples, whereas Figure 6a,b show the 1st and 2nd heating scans, respectively.

### 3.4. Physical and Mechanical Properties

Figure 7 shows the physical aspects of the polymer films as the 3M2MP fraction increased from 5.5 mol% to 100 mol%. Accordingly, the polymers became more transparent and softer as the 3M2MP content increased. Notably, the homopolymer tends to be very sticky, leading to handling difficulties.

Table 4 shows the mechanical properties based on the physical observations. As the 3M2MP fraction increased from Samples 1 to 7, the yield strength, tensile strength, and Young’s modulus decreased, whereas elongation at break increased. Sample 6 differed from the other samples, as explained above. The 3M2MP unit conferred the copolymers with great flexibility, allowing them to achieve over 1500% elongation at break with only 53.9 mol% of 3M2MP. Although Sample 5 showed similar 3M2MP contents to Sample 4, it exhibited a significantly lower elongation of break. This difference may be related to its lower molecular weight and differences in the degree of the block sequence (Table 2 and Table 3). Therefore, the sequence distribution may greatly affect the mechanical properties of the copolymers [37]. The stress–strain curves of all samples are shown in Figure 8.

The homopolymer also showed exceptional elasticity, with 2605% of elongation at break. Notably, most deformation was instantly recoverable. Figure 9 shows the elongation and recovery after manual deformation.

## 4. Discussion

NMR spectroscopy confirmed that P(3M2MP) homopolymer and copolymers with 3HB were biosynthesized by recombinant *E. coli* LSBJ using a similar method to previous studies to produce P(3H2MB) and its copolymers [19,38]. The biosynthesis data indicated that the maximum uptake of 3-mercapto-2-methylpropionic acid by the cells occurred under the culture conditions used in this study, unlike those observed for other sulfuric fatty acids [22,23,24]. Notably, the polymer content achieved for copolymer production was markedly higher than that observed for other P(3HB-*co*-3-mercaptoalkanoate)s [22,23,24], demonstrating that the point mutation D4N in PhaP_Ac_ and double-point mutations NSDG in PhaC_Ac_ lead to high polymer accumulation and high incorporation of 3M2MP unit into the 3HB copolymer. These findings indicate the relatively high polymerization activity of PhaC_Ac_ NSDG for 3M2MP unit. The lower production of the homopolymer relative to its 3HB copolymers suggests that the presence of 3HB unit facilitates the polymerization of 3M2MP unit by PhaC_Ac_ NSDG.

In contrast to P(3MP) and other bio-PTEs [25,39], P(3M2MP) can be easily dissolved in chloroform for extraction; however, the range of solvent solubility, which may impact its recovery from cells, remains unclear. Thakor et al. [40] efficiently extracted a PTE homopolymer from cells containing up to 45 wt% polymer by adding SDS to the cultivation broth, performing intensive stirring for 20 min at 90 °C, and conducting centrifugation and water rinsing cycles. This extraction method may also be applied to P(3M2MP).

Based on the sequence distribution, the *D* values of the copolymers were markedly higher than those typically found for most PHA copolymers [32,41,42] and other P(3HB-*co*-3-mercaptoalkanoate)s [26,43], indicating a high block sequence distribution. When the amount of 3M2MP precursor added to the medium was small, the degree of the block sequence tended to be high (Table 2) because polymerization of the 3HB monomer was prioritized over that of the 3M2MP monomer. Therefore, increasing the amount of 3M2MP precursor increased the intracellular 3M2MP concentration, which facilitated the polymerization of 3M2MP, thereby reducing the degree of block sequence in the biosynthesized copolymers. However, as the copolymers were not fractionated with a solvent depending on the solubility difference between the 3M2MP and 3HB units, some polymer samples may be blends of copolymers with different comonomer compositions.

The molecular weight of the copolymers was exceptionally high compared with that of normally biosynthesized P(3HB) but showed a decreasing trend with higher addition of 3M2MP precursor, which may induce a chain-transfer reaction during polymerization [34]. Moreover, regarding its thermal properties, in contrast to other bio-PTEs [25], P(3M2MP) exhibited amorphous behavior, possibly because a racemic 3M2MP precursor was used. Therefore, as the 3M2MP fraction in the copolymer increased, *T_g_* and *ΔH_m_* decreased. On the other hand, Huang et al. [44] demonstrated that the higher thioester linkage contents in 3HB-based copolymers may lead to higher thermal stability by reducing thermal degradation by hindering random scission of the polymer chain. Thus, thermal degradation study of P(3M2MP) and its copolymers are of interest as a potential target for thermostable material.

P(3M2MP) showed extraordinary elongation at break and instant recovery, indicating that its elasticity is higher than those of most commercial elastomeric polymers, such as natural or synthetic rubber (typically ranging from 100% to 800%) [45,46,47], or well-known elastic PHAs, such as poly(4-hydroxybutirate), which have copolymer and/or homopolymer elongations of approximately 1000% [48]. Because P(3HB*-co-*3MP) also has significant elastic properties [26], the superior elasticity of PTE may be related to the sulfur atom in the polymer backbone. Because the Pauling electronegativity of the sulfur atom (2.58) is markedly lower than that of the oxygen atom (3.44) and closer to the carbon atom (2.55) [25], the smaller difference in electronegativity may enable easier mobilization of the polymer structure during stretching [26]. α-Carbon methylation of 3M2MP may also affect its mechanical properties by hindering packing of the polymer structure.

Despite the lack of oxoester-equivalent polymers for analogy, a unique property of α-methylated monomers in their materials can be identified relative to those in other PTEs. It is interesting to study the biodegradability of P(3M2MP) because P(3MP), a non-α-methylated PTE, does not show biodegradability [49]. Bacterial PTEs are now considered non-biodegradable biopolymers [50]. Although the biodegradability of P(3MB) and P(3MV) has not been evaluated, other enzymatically polymerized PTEs were reported to be degradable with lipase [51].

## Figures and Tables

**Figure 1 bioengineering-09-00228-f001:**
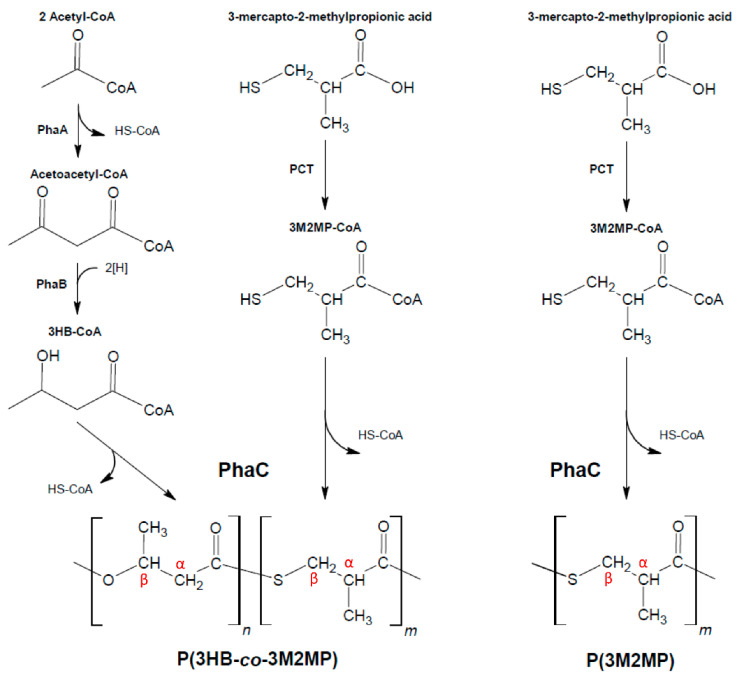
P(3HB-*co*-3M2MP) and P(3M2MP) biosynthetic pathway.

**Figure 2 bioengineering-09-00228-f002:**
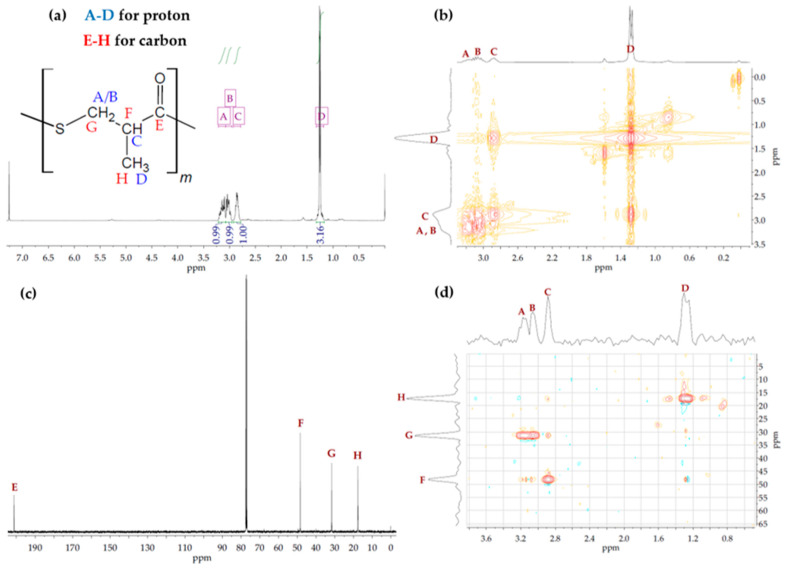
NMR spectra of P(3M2MP). (**a**) ^1^H NMR; (**b**) ^1^H-^1^H COSY NMR; (**c**) ^13^C NMR; and (**d**) ^1^H-^13^C HSQC NMR.

**Figure 3 bioengineering-09-00228-f003:**
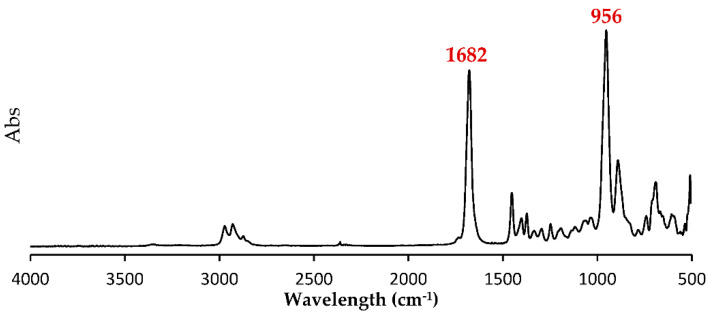
FTIR spectra of P(3M2MP).

**Figure 4 bioengineering-09-00228-f004:**
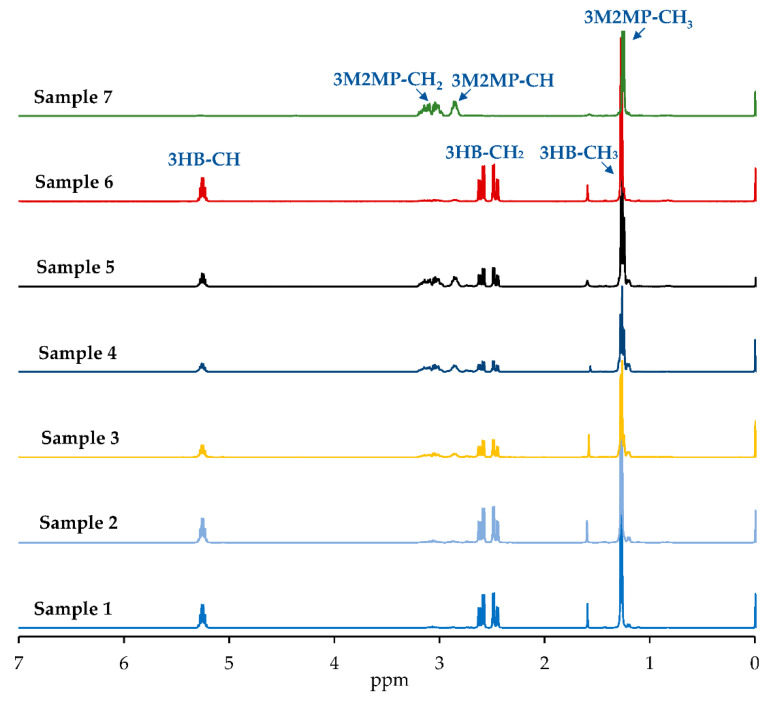
^1^H NMR spectra of Sample 1 (3M2MP 5.5 mol%), Sample 2 (10.1 mol%), Sample 3 (34.2 mol%), Sample 4 (53.9 mol%), Sample 5 (54.8 mol%), Sample 6 (10.7 mol%), and Sample 7 (100 mol%).

**Figure 5 bioengineering-09-00228-f005:**
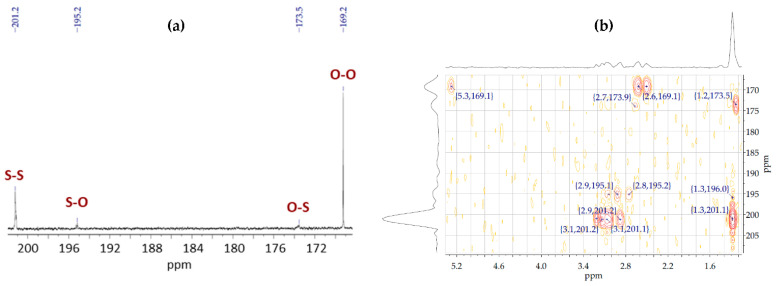
(**a**) ^13^C NMR of Sample 5 (3M2MP 54.8 mol%) showing combination peaks; (**b**) ^1^H-^13^C HMBC of Sample 5.

**Figure 6 bioengineering-09-00228-f006:**
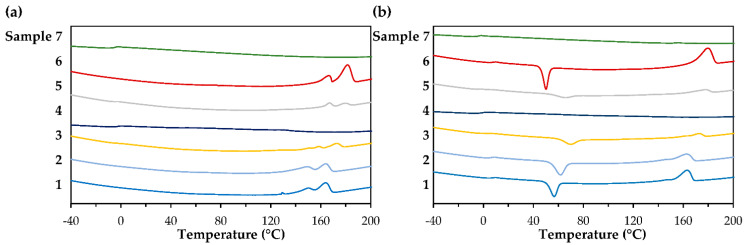
DSC thermogram of Samples 1 (3M2MP 5.5 mol%), 2 (10.1 mol%), 3 (34.2 mol%), 4 (53.9 mol%), 5 (54.8 mol%), 6 (10.7 mol%), and 7 (100 mol%). (**a**) 1st heating scan; (**b**) 2nd heating scan.

**Figure 7 bioengineering-09-00228-f007:**
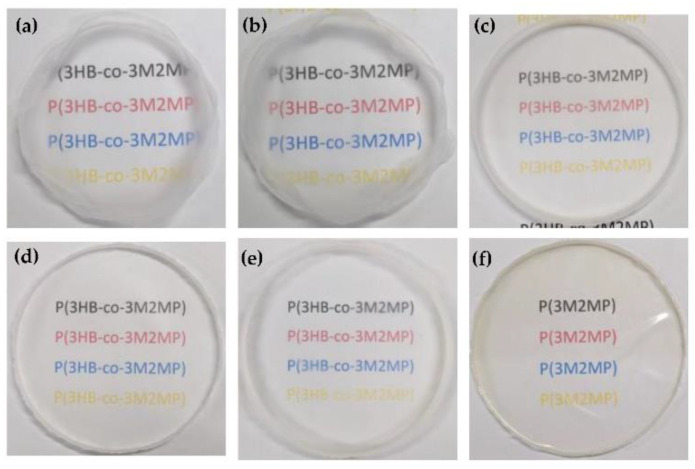
Polymer solvent cast films with increasing 3M2MP units. (**a**) Sample 1 (5.5 mol% 3M2MP); (**b**) Sample 2 (10.1 mol%); (**c**) Sample 3 (34.2 mol%); (**d**) Sample 4 (53.9 mol%); (**e**) Sample 5 (54.8 mol%); (**f**) Sample 7 (100 mol%).

**Figure 8 bioengineering-09-00228-f008:**
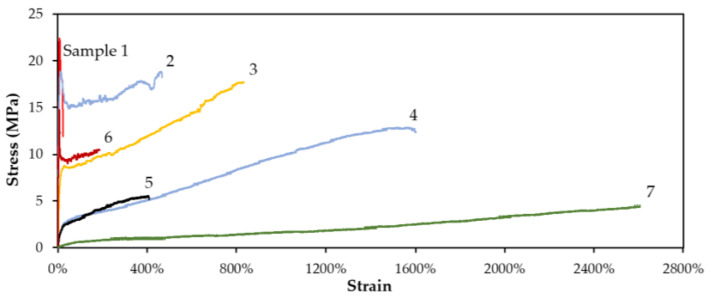
Stress–strain curve of Samples 1 (3M2MP 5.5 mol%), 2 (10.1 mol%), 3 (34.2 mol%), 4 (53.9 mol%), 5 (54.8 mol%), 6 (10.7 mol%), and 7 (100 mol%).

**Figure 9 bioengineering-09-00228-f009:**
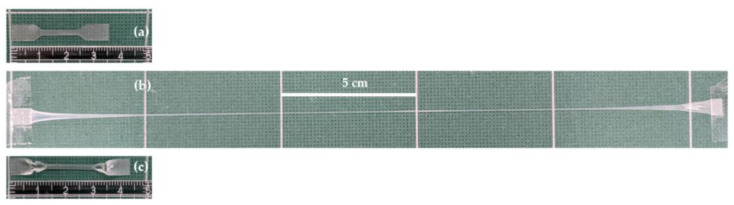
Elongation of P(3M2MP) (Sample 7) by manual deformation. (**a**) Before deformation; (**b**) during deformation; (**c**) after deformation.

**Table 1 bioengineering-09-00228-t001:** P(3HB-*co*-3M2MP) and P(3M2MP) biosynthesis by recombinant *E. coli* LSBJ.

PhaAB and PhaJ Expression	Precursor (g/L) ^1^	Dry Cell wt. (g/L)	Polymer Content (wt%)	Monomer Composition (mol%) ^2^		Sample ID
				3HB	3M2MP	
+ ^3^	0.25	3.50 ± 0.03	69.1 ± 1.9	94.5	5.5	1
+ ^3^	0.50	3.69 ± 0.25	68.7 ± 1.0	89.9	10.1	2
+ ^3^	1.00	3.48 ± 0.02	67.3 ± 0.1	65.8	34.2	3
+ ^3^	1.50	3.63 ± 0.02	69.1 ± 0.2	46.1	53.9	4
+ ^3^	2.00	4.29 ± 0.07	72.1 ± 0.8	45.2	54.8	5
+ ^3^	2.50	5.06 ± 0.04	77.2 ± 0.3	89.3	10.7	6
− ^4^	1.20	1.28 ± 0.01	8.4 ± 1.3	0	100	7

3HB, 3-hydroxyburyrate; 3M2MP, 3-mercapto-2-methylpropionate. Results are expressed as the mean ± standard deviation (*n* = 3). ^1^ (*R*,*S*)-3-Mercapto-2-methylpropionic acid. ^2^ Determined by comparison of the methine (>CH-) peaks in the ^1^H NMR spectra. ^3^ Strain harboring pTTQ19PCT and pBBR1phaP(D4N)JC_Ac_AB_Re_ NSDG. Cells were cultured in 100 mL M9-modified medium containing 0.25–2.5 g/L precursor (3-mercapto-2-methylpropionic acid), 20 g/L glucose, and 1 mM IPTG at 30 °C for 72 h. ^4^ Strain harboring pTTQ19PCT and pBBR1phaP(D4N)C_Ac_ NSDG. Cells were cultured in 100 mL M9-modified medium at 30 °C for 4 h, and 1.2 g/L precursor (3-mercapto-2-methylpropionic acid), 3.75 g/L glucose, and 1 mM IPTG were added before further culture for 72 h.

**Table 2 bioengineering-09-00228-t002:** *D* value and sequence distribution of P(3HB-*co*-3M2MP) samples.

Sample ID	3M2MP (mol%)	Diad Sequence Distribution				*D*	Degree of Block Sequence ^1^
		*F_OO_*	*F_OS_*	*F_SO_*	*F_SS_*		
1	5.5	0.960	0.016	0.009	0.015	100	High
2	10.1	0.934	0.012	0.006	0.048	623	High
3	34.2	0.741	0.006	0.037	0.216	720	High
4	53.9	0.329	0.182	0.152	0.337	4	Low
5	54.8	0.449	0.067	0.045	0.439	65	Medium
6	10.7	0.868	0.057	0.021	0.054	39	Medium

^1^ 1 < *D* < 10: low, 10 ≤ *D* < 100: medium, 100 ≤ *D*: high.

**Table 3 bioengineering-09-00228-t003:** Molecular weight and thermal properties of the samples.

Sample ID	3M2MP (mol%)	*M_w_* (×10^5^)	*M_w_*/*M_n_*	*T_g_* (°C)	*T_cc_* (°C)	*T_m_* (°C)	*ΔH**_m_* (J/g)
P(3HB) ^1^	0	5.2	2.3	4.0	NA	176	79
1	5.5	14.6	3.3	8.1	57	150, 164	28
2	10.1	12.0	3.2	6.6	62	149, 164	26
3	34.2	10.0	3.6	5.4	70	159, 173	13
4	53.9	13.7	4.4	−1.2	-	-	-
5	54.8	7.7	4.2	−3.1, 7.0	65	167, 179	9
6	10.7	17.5	5.6	8.4	50	166, 181	51
7	100	15.0	2.6	−3.1	-	-	-

NA: not available, -: not detectable. ^1^ Data from Ref. [19].

**Table 4 bioengineering-09-00228-t004:** Mechanical properties of the samples (*n* = 3).

Sample ID	3M2MP (mol%)	Yield Strength (MPa)	Tensile Strength (MPa)	Elongation at Break (%)	Young’s Modulus (MPa)
P(3HB) ^1^	0	NA ^2^	58 ± 7	12 ± 0	1420 ± 80
1	5.5	13.6 ± 2.0	23.0 ± 2.3	24 ± 13	880 ± 75
2	10.1	14.6 ± 2.2	19.3 ± 3.0	470 ± 131	644 ± 41
3	34.2	6.5 ± 0.3	17.1 ± 0.8	825 ± 21	87 ± 8
4	53.9	2.2 ± 0.1	12.7 ± 0.2	1549 ± 103	13 ± 1
5	54.8	1.3 ± 0.2	5.8 ± 0.4	436 ± 49	18 ± 1
6	10.7	8.2 ± 3.5	15.5 ± 1.4	158 ± 58	709 ± 303
7	100	1.0 ± 0.1	4.2 ± 0.5	2605 ± 4	0.8 ± 0.1

^1^ Ref. [19], ^2^ Not available.

## Data Availability

Not applicable.

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
