# Peer review of "Poly(3-mercapto-2-methylpropionate), a Novel α-Methylated Bio-Polythioester with Rubber-like Elasticity, and Its Copolymer with 3-hydroxybutyrate: Biosynthesis and Characterization"

_bioengineering, 2022, doi:10.3390/bioengineering9050228_

Round 1

Reviewer 1 Report

Dear Editor, in this study, a new polythioester (PTE), poly(3-mercapto-2-methylpropionate) [P(3M2MP)], from 3-mercapto-2-methylpropionic acid as a structurally-related precursor, and its copolymer with 3-hydroxybutyrate (3HB) were successfully biosynthesized and characterized. The paper is a completely study and contains new and interesting data. For this reason I propose to accepted for publication. Please find below some minor importance comments.

Why there are so big differences in mechanical properties, especially for elongation at break, in the studied samples?

Do the authors have any indication about thermal stability of these samples?

Author Response

Reviewer 1

Comments and Suggestions for Authors

Dear Editor, in this study, a new polythioester (PTE), poly(3-mercapto-2-methylpropionate) [P(3M2MP)], from 3-mercapto-2-methylpropionic acid as a structurally-related precursor, and its copolymer with 3-hydroxybutyrate (3HB) were successfully biosynthesized and characterized. The paper is a completely study and contains new and interesting data. For this reason I propose to accepted for publication. Please find below some minor importance comments.

A: Thanks for your positive comments to our paper. This revised manuscript was edited by a commercial proofreading service.

Q: Why there are so big differences in mechanical properties, especially for elongation at break, in the studied samples?

A: As summarized in Table 2, similar copolymer compositions have different degrees of block sequence. For example, samples 4 and 5 both have a 3M2MP fraction of around 54-55 mol%, but the degree of block sequence is significantly different. Sample 4 has a lower block sequence property and therefore a higher elongation at break. In addition, sample 4 has a high molecular weight. Due to these factors, the mechanical properties are different.

Q: Do the authors have any indication about thermal stability of these samples?

A: We plan to conduct study on thermal stability of this PTE in the future. However, it can be expected that α-methylation in 3M3MP can suppress the thermal decomposition of the main chain due to cis-elimination.

Reviewer 2 Report

The interesting study is well prepared with novel results, clear presentation and logical discussion. I only have the follwoing comment:

In methods section, please give more detailed information in assay the unit content of the 3M2MP in copolymers (In fig4, the methine siganal of 3M2MP seems weak in samples 2/3/6). 

Author Response

Reviewer 2

Comments and Suggestions for Authors

The interesting study is well prepared with novel results, clear presentation and logical discussion. I only have the following comment:

A: Thanks for your positive comments to our paper. This revised manuscript was edited by a commercial proofreading service.

Q: In methods section, please give more detailed information in assay the unit content of the 3M2MP in copolymers (In Fig4, the methine signal of 3M2MP seems weak in samples 2/3/6). 

As mentioned in the text, 3M2MP fraction was determined by comparison of the methine (>CH-) peaks in the 1H NMR spectra. The methine signal of 3M2MP in Figure 4 might be weak. This is because the sample concentration for NMR me